# Sustainable Water Treatment with Induced Bank Filtration

**Miles Schelling [1], Kavita Patil [2] and Thomas B. Boving [3,\*]**

1   The Aquaya Institute, P.O. Box 21862-00505, Nairobi 00100, Kenya
2   TERI—The Energy Research Institute, Western Regional Center, Alto-St. Cruz, Tiswadi 403202, India
3   Department of Geosciences & Department of Civil and Environmental Engineering, University of Rhode Island, 9 East Alumni Avenue, Kingston, RI 02881, USA
*   Correspondence: tboving@uri.edu

**Abstract:** This study demonstrates that an induced bank filter (IBF) system can treat raw water polluted with *Escherichia coli* (*E. coli*) bacteria. Similar to riverbank filtration (RBF), induced or reversed bank filtration relies on natural processes to clean water, including filtration through layers of allochthone alluvial sediments and a bioactive layer that forms on top of the filter after a ripening period. At the study site, located in Southwestern India, villagers rely on a mountain spring for their water supply. Although of generally high quality, the spring water contains *E. coli* bacteria (up to ~2000 MPN/100 mL). Raw water diverted from this spring was gravity-fed into the IBF system, which consisted of a (1) flow regulator, (2) pre-filter and (3) the actual IBF filter. Designed and constructed based on pilot testing of prototype filters, a full-scale filter (5 m by 7 m by 2 m) was built and its performance and maintenance requirements were studied during both the monsoon season and the dry season. The data show that the IBF significantly improved the water quality. Turbidity and *E. coli* concentrations were reduced to or below the detection limit (approximately 2.5 log unit reduction). During the peak of the monsoon season (August), *E. coli* was present in the IBF effluent after a storm destroyed the cover of the IBF tank. The IBF construction and maintenance costs were documented. Extrapolated over a 10-year period, the cost of IBF water was 3 and 10 times lower than reverse osmosis or water supplied by truck, respectively. This study demonstrates that IBF can be part of an affordable water supply system for rural villages in mountainous terrain where conventional RBF systems cannot be installed or where other water treatment technologies are out of financial reach.

**Keywords:** bank filtration; *E. coli*; water treatment; India; sustainability; rural community water

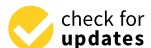



## 1. Introduction

Rural villages across India, as in many other developing communities around the globe, struggle to access clean water and, all too often, villagers are forced to drink water that is detrimental to their health [1]. A study of community water systems in the Western Ghats, India, showed that 80% of spring-fed water systems tested positive for *Escherichia coli* (*E. coli*) [2], which is an indicator for bacteria that can cause diarrhea and other gastrointestinal diseases. Because villages such as the one studied herein typically lack the financial and technical resources to improve their drinking water quality, research in sustainable and replicable water treatment options can provide a framework for continued progress toward improving health in rural Indian villages [3].

One sustainable and replicable water treatment option that has been successfully applied for treating polluted water in South India is riverbank filtration (RBF) [4]. This technology involves drilling one or more wells near a river and hydraulically pulling water through the alluvial bank sediments to attenuate contamination [5]. As polluted river water passes through the alluvial sediments, it is cleaned by a number of naturally occurring biological, physical and chemical processes, such as the predation of bacteria in a collimation layer at the contact of the river water and the sediment [6], as well as straining processes [7] or chemical transformation reactions that take place during the passage of the

water through the alluvial sediments [8]. Which of these treatment processes dominates largely depends on the contaminant type [9]. Importantly, the treatment processes within the collimation layer are regenerative because periodic scouring during flood events naturally regenerates its treatment activity [10]. This ensures that the RBF system can remain effective almost indefinitely. Finally, because a properly designed RBF well attracts mostly riverbank filtrate and only a fraction of groundwater [11], this technology reduces the pressure on already stressed aquifers. This is an important consideration when developing new water resources in areas where groundwater tables are dropping due to over-pumping, as in many rural regions of India [12].

The naturally occurring water filtration processes that allow conventional RBF systems to function were recreated in this project but under reversed flow conditions. In other words, surface water enters the alluvial sediments and flows through the layers forced by gravity instead of pumping. As such, we refer to this treatment approach as induced bank filtration (IBF). The mechanisms of filtration in IBF mimic those in RBF and slow sand filters. Straining of suspended solids, including the contaminants associated with those solids, is the most effective treatment process in the topmost layer of the IBF, where particles are retained that are too large (approximately 0.155 times the sand grain diameter [13]) to fit through the pore spaces [14]. Smaller particles not strained in the top layer are attenuated through sedimentation, whereas diffusion, sorption, electrostatic and electrokinetic interactions deeper within the filter matrix attenuate dissolved compounds [13].

The top layer of the IBF system is colonialized by benign and beneficial microbes present in the raw water [15]. These microbes form a bioactive collimation layer, which becomes an essential component of the filter, i.e., most of the pathogenic bacteria removal occurs in this biolayer [11]. In RBF and slow sand filter systems, it takes an approximately two-week ripening period for this layer to form before optimal performance is achieved [16]. A similar ripening period must be expected for an IBF. Inside the collimation layer, which typically is less than 2 cm thick [17], organic matter present in the raw water is gradually broken down and converted into water, $CO_2$ and relatively inoffensive inorganic salts, such as sulphates, nitrates and phosphates [15]. The bioactive but benign microorganisms in the collimation layer predate on harmful bacteria, such as *E. coli* [11], and kill them by excreting toxins [15].

Over time, accumulating particles form a caked layer at the very surface of the filter and eventually create resistance to flow and cause head loss [14]. Bouwer et al. (2000) showed that clogging of the top layer can cause a 95–99% reduction in the hydraulic conductivity of sand filters [18]. In consequence, an IBF system likely requires some maintenance in the form of scraping the top layer of sediment, backwashing or by systematically re-sanding, where sand is removed from the filter, thoroughly washed and returned to the filter [8,19–21]. Cleaning by scraping during IBF maintenance will destroy the collimation layer and it will require a few days to regenerate before optimal performance is resumed [15,19]. The need for maintenance is a potential drawback of IBF relative to RBF, where strained fine materials are periodically removed during naturally occurring floods. Therefore, an investigation of IBF maintenance requirements was part of this study.

Arguably, IBF could be considered a form of designed slow sand filters and as such is not a novel idea. Slow sand filters are used throughout the world to treat water from household to city scales [22]. However, unlike most slow sand filters, the design of an IBF system is guided by the alluvial sedimentology of the local riverbanks. In other words, in constructing an IBF, only locally sourced alluvial materials are used and layered to mimic local riverbank systems as closely as possible. The promise of this approach is threefold: (1) the filter material can be inexpensively sourced locally in most locations; (2) the geological composition of the filter material is identical to the material coming into contact with the local surface and groundwater, which ensures that the olfactory properties of the filtrate will be very similar to traditional water sources. The taste of water is an official water quality parameter [23] and an important consideration when supplying treated water to communities in India [24]. Finally, (3) beneficial bacterial communities that make up

the culmination layer on top of the IBF are likely not different from those present in local rivers, i.e., the interplay of filter material mineralogy, water chemistry and bioactivity will not change to a great degree, as may happen when allochthonous material is used for filter construction. This facet of IBF is expected to support bacteria adapted to local conditions and will ensure that naturally occurring biological filtration processes are as effective and sustainable as possible. We, therefore, hypothesize that these characteristics of IBF permit the construction of a comparatively inexpensive water treatment system that will produce water that is acceptable, by taste, to the local community and which meets local drinking water standards.

In the following, we summarize the results of the first systematic investigation of an IBF water treatment system in this part of India. We compare the pre-intervention water quality with our post-intervention data. We also present the hydrogeological properties of locally sourced alluvial sand and gravel deposits. We then compare the results of both pilot- and full-scale filter tests and examine the effectiveness of the IBF toward improving the water quality. The quantifiable measures for these outcomes are water quality data in terms of *E. coli* and total coliform bacteria concentrations before and after IBF and periodic field testing of turbidity as well as inorganic (e.g., major ions) and organic (e.g., pesticides) parameters, besides others. A brief analysis of the cost of constructing and maintaining the IBF system is also included. This project adds to the knowledge base of affordable water treatment techniques in developing nations and will provide valuable information for future water development projects in this part of India.

## 2. Methods

### 2.1. Study Area

The study site is the village of Nersa located in Southwestern India (Figure 1)—specifically, the Khanapur Taluk of Belgaum District, Karnataka (15°36′00.26″ N, 74°25′39.92″ E). The village lies on the western edge of the Western Ghats Mountain range, which traverses north/south along the coast of India. Nersa's elevation is approximately 700 m above sea level. The entire area is within the Krishna River Basin. The climate is sub-tropical. The mean annual rainfall is 1859 mm, of which 72% occurs during the monsoon from June to September [25]. Precipitation data were obtained from rain gauges at a field station in Belgaum, 40 km from the study site [26]. Agriculture is the main source of employment for the people of Nersa.

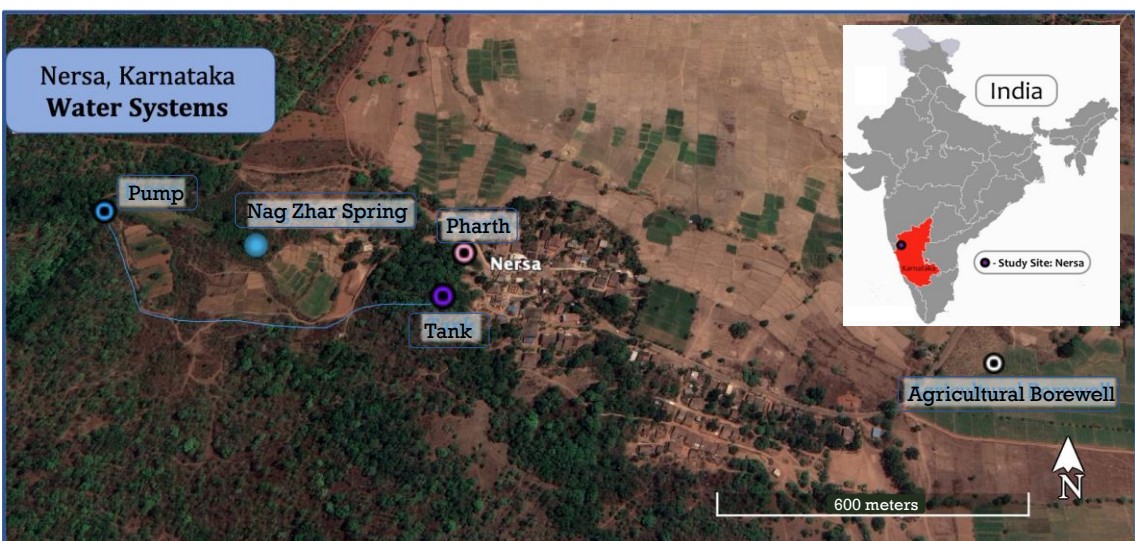

**Figure 1.** The study area is located in the Khanapur Taluk of Belgaum District, Karnataka in Southwestern India.

The area around Nersa is classified as rugged terrain with bedrock of schist and basalt. The soil types are shallow to very deep black soils, red loamy soils and lateritic soils. The major water-bearing formations are gneiss, schists, limestones, sandstones, basalts and alluvium. Most springs in this region originate between massive and vesicular units of basalt [27]. Das et al., 2005 provide insights into the chemical composition of groundwater in the Western Ghats region [28]. In a typical year, the depth to water level was 1.25 to 11.80 m before the monsoon and 0.85 to 9.50 m after the monsoon. Exploratory wells in the region yielded 0.02 to 7.58 L per second and drawdown ranged from 0.1 to 32.4 m, with transmissivity between 1 and 2220 m$^2$/day [25].

The village's current water supply originates at a makeshift dam catching water from two rheocrine springs running along a streambed. The dam is roughly 10 m long and 1 m high, holding approximately 30 m$^3$ of water. Each year, this dam must be rebuilt after being destroyed by the monsoon. Spring water caught behind the dam flows into a 3″ (7.6 cm) pipe, which travels 3.5 km, with an elevation change of 30 m, to the village of Nersa, where it terminates in an open channel of a few meters in length before entering the main collection point, named a Pharth. This area holds significant spiritual significance for the villagers; certain customs and regulations govern the space. For example, shoes are to be removed upon approaching and while gathering water. During the dry season, the flow at the Pharth is around 50 L/min, with fluctuations caused by rain events and damages or repairs to the pipeline. Maintenance of the dam and pipeline is completed by volunteers. Villagers complain that the Pharth will often go dry during the months of April and May and they will have to gather water from agricultural open wells.

A reverse osmosis (RO) system was constructed in the village in 2005. It was not maintained properly and quickly became inactive. While it was operational, villagers paid 2 rupees for 10 L of water. Only some families used the RO and many complained that it had a poor, oily taste.

### 2.2. Water Testing Methods

Water quality baseline surveys were carried out in 2019 and 2020. One water testing location was the Pharth—the village's principle drinking water source. Other sampling locations included a low-output spring, named Nag Zhar, located downgradient from the village, and an agricultural bore well with a water depth of around 8 m, and a household water tank supplied by Pharth water (Figure 1). Samples were collected in 5 and 10 L plastic jugs and sent to a commercial laboratory (VIMTA in Pune, India) on the same day. They were tested for major ions, heavy metals, parasites and pesticides (Table S1). Handheld devices were used to test for pH (Middons, digital pH meter), electrical conductivity (EC) and total dissolved solids (TDS; Sumgot TDS meter) and temperature, as well as field test kits for nitrate, alkalinity and phosphate (LaMotte, 2018). *E. coli* and total coliform samples were collected in 100 mL sterile plastic vials and stored in an ice-cooled freezer box while being transported to Panjim, Goa, and analyzed by the Energy and Resources Institute (TERI, Panjim, India) laboratory, a partner in this project. *E. coli* bacteria are an indicator of fecal pollution and their absence in any 100 mL of water is required by the Indian drinking water standards (BIS 2012). Further, the Indian drinking water standards consider total coliform bacteria not an acceptable indicator of the sanitary quality of rural water supplies, particularly in tropical areas, where many bacteria of no sanitary significance occur in almost all untreated supplies. However, total coliform results are reported herein for comparison with other regions where this water quality indicator might be applicable. The bacteriological water samples were analyzed using the IDEXX method within 8 h of collection [29,30].

After extensive conversations with the villagers about their expectations and needs in terms of an acceptable water supply system, a full-scale IBF system was constructed during the 2020 field season (January through March). The season was suspended in March 2020 due to the COVID-19 pandemic and subsequent lockdown. An opportunistic single-event IBF influent and effluent sample was collected in February 2021. The field season resumed

in January 2022 and ended in October 2022. It covered the monsoon rainy season and the dry period before and after it. During this time, the IBF filter system's performance and maintenance requirements were evaluated.

### 2.3. IBF Construction and Testing

In preparation for building a full-scale IBF from local materials, alluvial deposits were mapped along nearby rivers, including the Bandura, Malaprabha and Mahadayi. Samples were collected from each location and characterized by sieving and porosity, following standard methods. Two experimental IBF filters were constructed using 300 L plastic tanks. Each tank was packed with unsorted alluvial material; one tank contained a layer of locally sourced granular activated carbon (GAC) and one without GAC. The performance of the two pilot-scale IBFs, including tracer test data, informed the construction of the full-scale IBF. The pilot-test data were summarized in Schelling et al. (2021) [31].

The full-scale IBF system consisted of a flow regulating tank, an ascending flow pre-filter and the actual IBF filter (Figure 2). The dimensions of the IBF filter were 5 m by 7 m by 2 m (W × L × H). It was constructed of reinforced concrete with an open top and a wall thickness of 15 cm. A sealant was used to prevent leakage in case of cracking. The tank floor sloped toward a 6.4-cm-thick PVC drainage pipe, which was connected to an underdrain constructed of 5.1 cm unplasticized polyvinyl chloride (UPVC) pipe with 10 mm holes drilled every 5.1 cm along the bottom of each pipe. The underdrain pipes were elevated 6.5 cm above the bottom of the tank. Local alluvial sand and gravel were packed in layers 1 m thick. Before installation, the raw material was separated into three size fractions by sieving and washed in a nearby river to remove debris and fine materials. The clean material was loaded by hand into the filter, with the larger-sized gravel installed first and fully covering the underdrain. Care was taken not to damage the PVC underdrain pipes. Each layer was raked flat before adding another layer. The top of the tank was covered by a plastic mesh and corrugated iron sheets to keep out animals and debris.

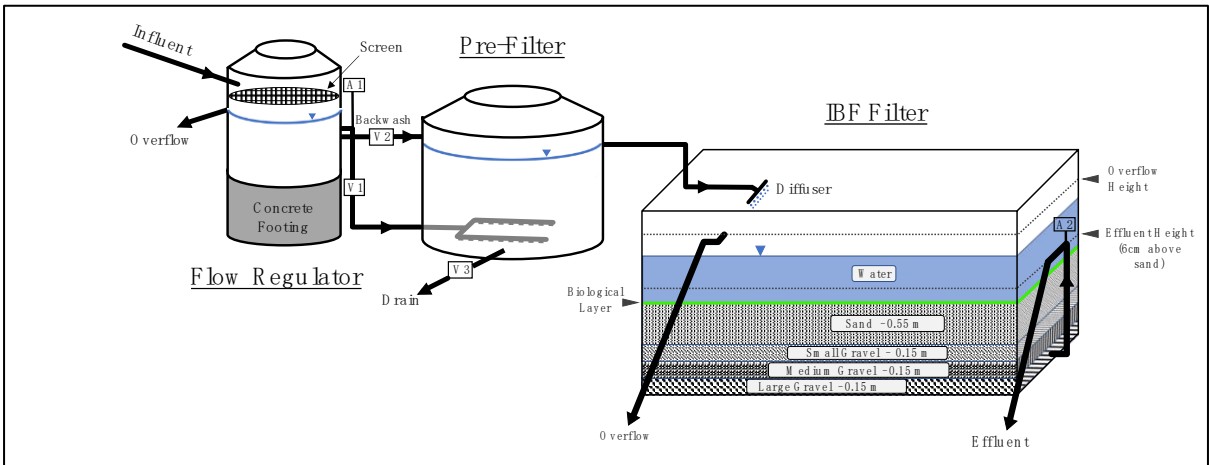

**Figure 2.** The induced bank filtration (IBF) treatment system consists of a flow regulator tank, ascending flow gravel pre-filter and descending flow IBF filter. During normal operation, valve V1 is open and valves V2 and V3 remain fully closed. A1 and A2 represent airlocks. Influent water samples were collected before the flow regulating tank. Effluent samples were collected after the IBF filter. Drawn not to scale.

Water supplied to the IBF system was diverted from the main pipeline that transmitted the raw spring water to the village. Any spring water not diverted to the IBF (approximately 20% of total flow, depending on the season) continued to flow to the village. Preserving some flow of unfiltered water was considered important because of the spiritual significance of the traditional Pharth water system. The diverted water was supplied by gravity flow to the IBF system via a 2.5 cm PVC pipe.

Raw water entered the elevated 300 L flow regulator tank (total difference in head: 2 m). Inflow first passed through a plastic mesh screen to remove coarse debris, such as leaves. The top of the tank opened to remove accumulated debris by hand. An overflow pipe located 8 cm above the tank outflow returned excess water to the Pharth. The head inside the flow regulator tank was maintained at 7 cm by valve V1 to ensure a constant flow, which created favorable conditions for the biolayer to form on top of the IBF [8,20]. Water leaving the flow regulator tank entered a gravel-filled pre-filter tank from the bottom through a 2.5 cm UPVC pipe with 5 mm holes drilled every inch, resulting in ascending flow conditions. Pre-filters have been shown to protect the lifespan of the main filter and can significantly contribute to bacteria removal during the ripening period of the biolayer forming on the main filter [32]. The pre-filter consisted of a 1000 L plastic tank filled with medium-sized gravel. Water ascended through the filter media and then drained via a 2.5 cm UPVC pipe leading from the top of the pre-filter to the actual IBF filter. Airlock A1 stopped air from entering the pre-filter. The pre-filter was backwashed by opening valve V2, allowing water to flow directly from the flow regulating tank into the top of the filter. Valve V3 allowed the draining and backwashing of the tank.

The water exiting from the pre-filter was gravity-fed to the actual IBF. Inflow first passed through a diffuser consisting of a horizontal pipe with 5 mm holes drilled every 2.5 cm. The diffuser reduced disturbances of the biolayer that formed on top of the IBF. Treated water exited the IBF through an underdrain connected to a 2.5 cm UPVC pipe raising to a height 8 cm above the top sand layer. This ensured that the sand and biolayer inside the tank were always submerged, even if influent water was to be cut off. A dipstick was used to measure the water height in the tank, which was expected to increase over time as biofilm formation and potential clogging reduce infiltration rates. Airlock A2 ensured that the IBF did not drain from suction. The filtered water exiting the filter traveled 10 m through a buried 2.5 cm UPVC pipe to the distribution point, where it continuously flowed from a concrete distribution point located adjacent to the traditional Pharth. The filter was continuously running at 6.7 L/min during the entire testing period (Figure 3).

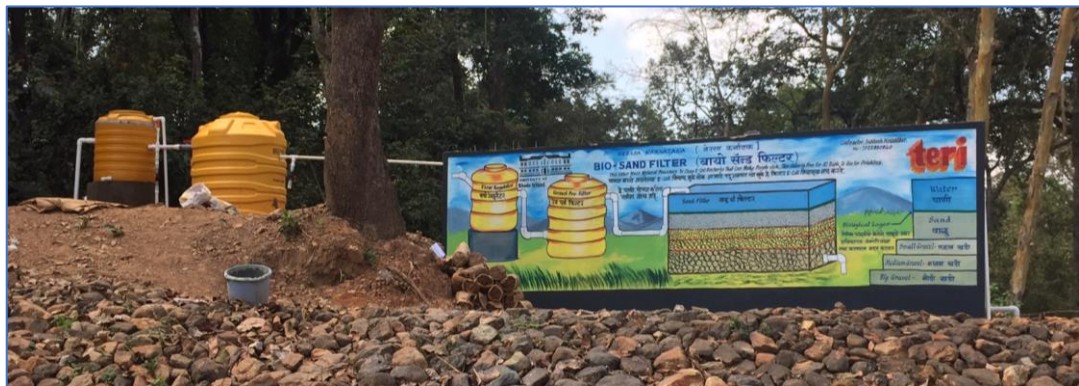

**Figure 3.** Picture of the IBF filter system. A diagram was painted on the filter wall giving a simple explanation of the system in both English and Marathi, the local language.

## 3. Results

In 2019, existing water sources were tested to establish a water quality baseline. These sources included the Pharth, a tap fed by Pharth water, an agricultural bore well and water from a spring downstream from the village (Nag Zhar spring). All water sources were free of detectable pesticides and parasites. Results for heavy metals were all non-detected, besides 0.002 mg/L chromium found in the bore well at 0.03 mg/L. All parameters, except *E. coli*, were below Indian standard BIS IS:10500 requirements (see SI). *E. coli* and total coliform were detected in all water sources, except the bore well. Electrical conductivity (EC) during the dry period was comparatively low and ranged between 19 and 68 μS/cm. EC readings during the monsoon were even lower (average: 21 μS/cm). The pH (5.1 to 5.5)

and the water temperature varied over a narrow range (21.4–26.0 °C). The Pharth water was sampled eight times for *E. coli* and total coliform between January 2019 and February 2020 (Figure 4). The data indicate that the concentrations of both bacteriological parameters dropped during the monsoon (June to August) but reached a maximum soon after the end of the monsoon, i.e., in October, when *E. coli* and total coliform reached 1509 MPN/100 mL and 61,300 MPN/100 mL, respectively. Two rain events in the dry season (March and April 2019) also corresponded with higher bacterial contamination.

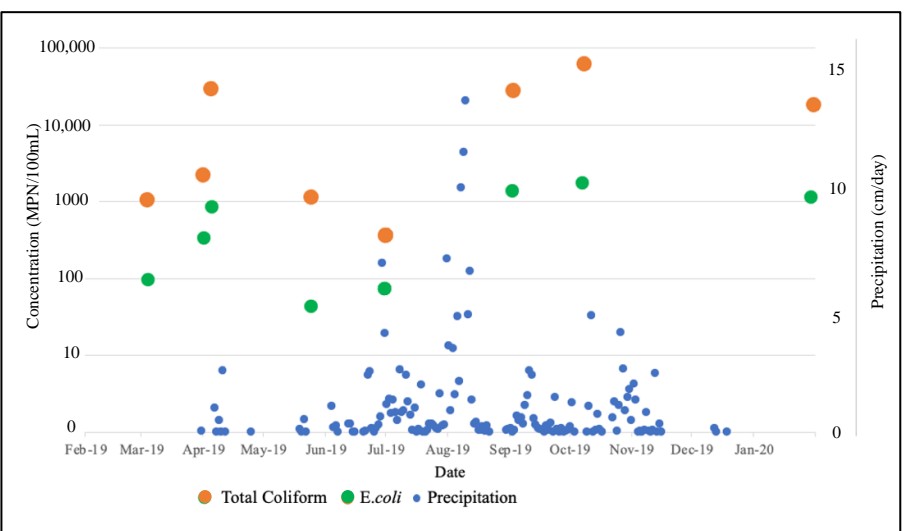

**Figure 4.** Time series of *E. coli* and total coliform in untreated water collected at the village's main water source (Pharth). Precipitation data from rain gauges at Belgaum Sambra field station [26].

*Full-Scale IBF System*

The IBF filter system was rinsed before the actual testing period to remove most of the finest materials. Afterward, the turbidity in the pre-filter effluent remained consistently at the detection limit, i.e., lower than the raw water entering it, which was as high as 4.5 NTU in March 2020. The turbidity of the water flowing from the IBF reached as high as 16.5 NTU during the initial month of testing but then gradually declined to a non-detectable level for the remainder of the test. The EC of both the pre-filter and IBF filter effluent was similar to that of the influent water and never exceeded 40 μS/cm (data not shown).

After the completion of the filter's construction in February 2020, the treatment effectiveness of the filter system was evaluated by comparing the raw water quality entering the pre-filter against the water leaving the IBF. As all parameters except bacteria were below the detection limit in the raw spring water, we did not test the effluent for all parameters. The treated water was clear and the quantity was sufficient to meet the demands of the villagers. The taste and odor of the water are parameters of the Indian Standard Specifications for Drinking Water (BIS 2012). Both of these parameters met the acceptable limits, i.e., they were agreeable to the villagers. Further, the pH and TDS of the treated water were well within the acceptable range (pH 6.5–8.5 and <500 mg/L, respectively). Because of the difficulty in transporting water samples during the COVID-19 pandemic, the testing for *E. coli* and total coliforms was performed sporadically. However, samples collected from the filter system at the end of the 2020 field season suggest that the filter quickly became effective in attenuating *E. coli* but had not reached its full potential, as indicated by the continued presence of elevated total coliforms (data not shown). During much of 2021, the IBF filter system was used by the villagers of Nersa until a section of the pipeline delivering raw water to the filter was destroyed during the monsoon. In early 2022, the affected pipeline section was fixed and buried 0.5 m deep to prevent future disruptions. Continuous flow to the IBF system resumed in late January.

Over the entire testing period (January to October 2022), the IBF removed on average 96.96% of *E. coli* and 97.46% of total coliforms. Greater than 99% removal (~2.5 log units) was observed after the biolayer on the IBF was fully developed. This improvement is illustrated in Figure 5A,B, which show that the bacteria removal was lower than average during the initial three weeks of operation, when the biolayer on top of the IBF filter had not fully developed. Once formed, the *E. coli* concentrations in the filter effluent (Figure 5A) remained at or below the detection limit (1 MPN/100 mL), except in August, when, during the height of the monsoon rains, 40 MPN/100 mL were detected in the effluent from the filter systems (versus 1320 MPN/100 mL in the influent). Similar results were observed for total coliform bacteria. In other words, once the biofilm matured, an approximately three to four orders of magnitude decrease in total coliform concentration was recorded (Figure 5B). However, removal during the August 2022 monsoon was less efficient, when the influent concentration only dropped by two orders of magnitude, from $1.2 \times 10^5$ MPN/100 mL to $1.2 \times 10^3$ MPN/100 mL. This drop in efficiency was likely due to damage that occurred to the cover of the filter during heavy rain and wind, causing subsequent contamination and disturbance of the biolayer (i.e., rain was able to fall directly into the filter and the biolayer was disturbed by broken cover material). A longer study period would be needed to determine if this episodic reduction in performance is typical during the monsoon.

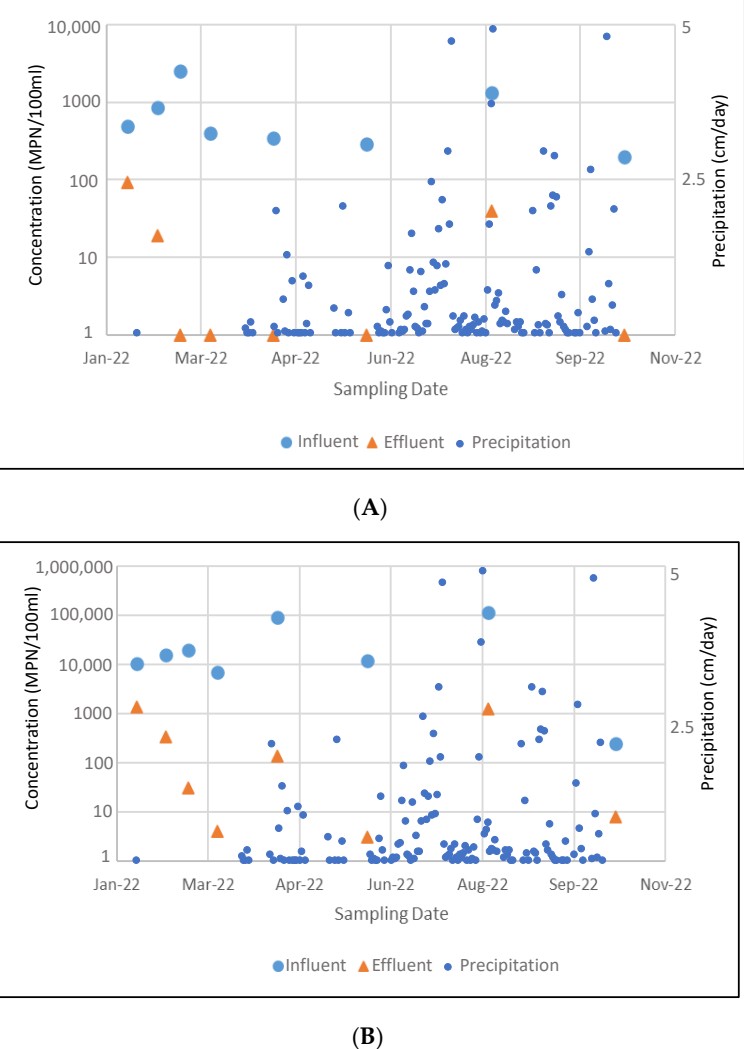

**Figure 5.** (**A**) *E. coli* and (**B**) total coliform bacteria measured in the influent water and the effluent from the IBF system. *E. coli* non-detected values were set to one to permit plotting on log scale. Precipitation data from rain gauges at Belgaum Sambra field station [26].

The cost of constructing the IBF filter system was USD 4600, of which USD 2100 was expended on labor (Table 1). The remainder covered locally sourced construction materials and construction machinery rental fees. The land on which the filter system was built was community-owned and land access was provided as an in-kind contribution. Villagers were trained to operate and maintain the filter system. This work was considered a community service and was provided without a fee. Because the filter system is gravity-fed and contains no electrical components, there are no costs associated with electrical power. All system hardware, such as pipes, valves and tanks, were assumed to last for at least 10 years, excluding catastrophic failures.

**Table 1.** IBF filter system costs in 2020 (USD).

| Cost Category | # Items | Cost Per Item | Total Cost (USD) |
| --- | --- | --- | --- |
| 1000 L plastic tank (pre-filter) | 1 | 60 | 60 |
| PVC pipe | 90 | 5 | 450 |
| Valves | 70 | 2 | 140 |
| Purchased sand/gravel filter material | 1 truck load | 350 | 350 |
| Concrete | 150 bags | 5 | 750 |
| Steel rebar (6 mm and 10 mm) | 250 | 1 | 250 |
| Rental of construction machinery | 2 days | 150 | 300 |
| Other construction material | NA | NA | 200 |
| Labor | 350 days | 6 | 2100 |
| **Total** | | | **4600** |

Consultations with villagers indicate that the average daily amount of water used for drinking and cooking is 10 L per person. Assuming that the filter runs for 10 years, with all 300 villagers using it as their primary water source, the IBF system would provide 10,950 $m^3$ of drinking water. This indicates a cost of USD 0.42 per $m^3$ of water. For comparison, when the RO was functional, villagers paid 1 rupee for 10 L (USD 1.3 per $m^3$). Water supplied by a tanker truck would cost approximately USD 4 per $m^3$.

## 4. Discussion

Wells and springs are the main water sources in the study area. The baseline survey of these sources, particularly the groundwater's low EC, suggests that the monsoon rains are the principal source of recharge to the groundwater in the study. The village's main water supply (Pharth) delivers water of generally high quality, except for occasional spikes in turbidity and bacteria concentrations. These spikes are most prominent after rain events outside and during the monsoon season. During the monsoon and relative to the dry season, both higher and lower bacteria concentrations are observed. Higher than average bacterial concentrations are reported for other monsoon-influenced parts of India [33–35], whereas Vincy et al. (2017) note that in the comparably sparsely populated Western Ghats, waters have comparatively low bacterial counts during the monsoon [36].

Prior to the construction of the full-scale induced bank filter systems, two filter designs (with and without GAC) were evaluated at the pilot scale. The results, published by Schelling et al., 2021, suggest that there is no apparent value in adding this particular locally sourced GAC material to the filter system [31].

The full-scale IBF system consisted of a flow regulator tank, a pre-filter and the actual IBF. The pre-filter's primary function is to remove fine particles and prevent other debris from entering the actual IBF. As measured by turbidity, it performed this task as expected. It is noted that forcefully rinsing the filter system before the actual testing period was critical to achieving the successful removal of turbidity during regular operation. Further, the observed similarity of the influent and effluent water in terms of EC after the initial two months of operation indicates that the IBF system had, as desired, little effect on the concentration of dissolved solids. This increase in EC by 200% in the water filter effluent would not have been unusual [37]. Lastly, the IBF filter system's *E. coli* removal performance

met expectations. In other words, after a fully mature biolayer formed on top of the IBF after 3 to 4 weeks of operation, bacteria concentrations in the filter effluent were at or below the detection limit. Removal of total coliforms was similarly effective. Lower removal was observed during the height of the monsoon season after a storm destroyed the cover of the IBF filter. Supported by anecdotal evidence, heavy rains appeared to have flushed pollutants, such as bird droppings, into the IBF tank. A sturdy, impermeable cover would likely alleviate this problem.

After extrapolation of the data from the two pilot-scale filters, and considering the volume (40 m$^3$), bulk porosity (39%) and design flow rate of the IBF filter (6.7 L/min), the water's residence time in the IBF filter was estimated to be 19 h. This residence time is within the range of other similar-sized water filters [13].

Informal interviews of the villagers who relied on the water from the newly constructed IBF filter indicated that the water was of acceptable quality in terms of taste and odor, compared to the traditional Pharth. Moreover, villagers reported that the IBF water caused less gastrointestinal illness, if any. However, it would take a detailed and systematic health survey to quantify the new filter's effects on the physical well-being of the villagers. Finally, a comparison of the costs of constructing and operating the IBF filter system relative to other water treatment approaches used in the study area showed that the IBF was 3 and 10 times more cost-efficient than RO or water supplied by truck, respectively. The cost assessment assumed the lifespan of the IBF to be 10 years.

## 5. Conclusions

This field study focused on demonstrating a cost-effective, sustainable and replicable solution to the water quality issues facing villages in the Western Ghats of India. After initial water quality testing and extensive conversations with local villagers about their needs and expectations, two pilot-induced bank filters (IBFs) were constructed and tested, leading to the installation of a full-scale IBF system. The necessary filter materials were easily sourced from local alluvial sediment deposits. The use of autochthonous filter material ensured that the treated water had similar olfactory properties compared to traditional water sources, which ensured the acceptance of the treated water by the villagers. The filter effectively removed bacteria and turbidity, with an approximately 2.5 log unit reduction. In addition, the improved water's perceived health effects were an important determinant of the successful adoption of the IBF technology in Nersa. The new water treatment system now provides clean water with a constant flow rate of 6.7 L/min, at a comparably low cost to this village, and may therefore serve as a model for other, similar villages that cannot afford alternative water treatment technologies.

**Supplementary Materials:** The following are available online at https://www.mdpi.com/article/10.3390/w15020361/s1, Table S1: Laboratory results from tests performed 24 February 2019.

**Author Contributions:** Conceptualization, T.B.B. and M.S.; Methodology, T.B.B., K.P. and M.S.; Validation, T.B.B., K.P. and M.S.; Resources, T.B.B., K.P. and M.S.; Data Curation, T.B.B. and M.S.; Writing—T.B.B. and M.S.; Writing—Review and Editing, T.B.B., K.P. and M.S.; Visualization, T.B.B., K.P. and M.S.; Supervision, T.B.B. and K.P.; Project Administration, T.B.B. and K.P.; Funding Acquisition, T.B.B. and M.S. All authors have read and agreed to the published version of the manuscript.

**Funding:** This research was funded by the InterExchange Christianson Fellowship, The Pollinator Project, Ocean State Job Lot/University of Rhode Island Presidents Fund Grant, University of Rhode Island Graduate Union Award.

**Data Availability Statement:** Data available upon request.

**Conflicts of Interest:** The authors declare no conflict of interest.

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
