# Peer review of "Sustainable Water Treatment with Induced Bank Filtration"

_water, doi:10.3390/w15020361_

Round 1

Reviewer 1 Report

This study is interesting but there are some shortcomings below in the paper. 

- The literature review could be enhanced. the references below should be added related to reverse osmosis:

* (2020). Grey water footprint assessment for a dye industry wastewater treatment plant using Monte Carlo simulation: influence of reuse on minimisation of the GWF. International Journal of Global Warming21(2), 199-213.

*(2019). Grey water footprint of a dairy industry wastewater treatment plant: a comparative study. Water Practice and Technology14(1), 137-144.

-In Table 1, a cost assessment has been carried out. Please give a method for calculating it. Also, a correlation between costs and treatment efficiency could be given. The reference below should be used:

* (2020). Energy cost assessment of a dairy industry wastewater treatment plant. Environmental Monitoring and Assessment192(8), 1-17.

** (2022). Investigating energy costs for a wastewater treatment plant in a meat processing industry regarding water-energy nexus. Environmental Science and Pollution Research29(1), 1301-1313.

- The conclusion should contain numerical findings. Please revise it.

- The language of the paper could be reviewed carefully. There are several grammer mistakes.

Reviewer 2 Report

This paper can be accepted after the authors modify these points:-

1. The authors should add a real photo of the IBF system.

2- The authors should add the chemical composition of the filter's materials to the revised manuscript.  

3- The authors should add real photos for the water before and after using the IBF filter.

4- The authors should add the mechanism of bacteria removal by this filter.

Reviewer 3 Report

This manuscript could be considered for publication by Water after a major revision considering the comments below: 

Abstract:

The authors start the abstract with “this field study demonstrates that…” Which field? This sentence should be rewritten.

Write out “Escherichia coli”, the first time this genus appears in the text. From the second time, you can write just “E. coli”.

Introduction

Always write scientific names in italic form (e.g. Escherichia coli in line 37)

Lines 37~38 “bacteria known to cause diarrhea and 37 other gastro-intestinal diseases“ : E. coli  is not necessarily patogenic, but an indicator of fecal contamination.

Line 43: “ Boving, 2017” is incorrectly cited.

Line 69: “Huisman 1974” is incorrectly cited.

Line 60: “Howe, 2012” is incorrectly cited.

Line 79: “Lynn, 2013” is incorrectly cited.

Line 85: “Howe, 2012” incorrectly cited.

Line 85: “Bouwer (2000)” should be “Bouwer et al”

Other places where the citation should be corrected: Line 90; Line 98

Line 112 “bacteria adopted to local conditions”: please check where it should really be “adopted” instead of “adapted”.

Methods:

Lines142-143 – agriculture is the main employer of what?

Line 190: “EPA, 2010” is not listed in the References section.

Figure 2: I highly recommend to improve the image quality. Also, the caption mention “A2”, but I could not find it in the figure.

References

According to the “instructions for authors”: “References must be numbered in order of appearance in the text“.

Table in Supplementary file:

5th line: “total dissolved” what???

Last column: what is the meaning of “o relaxatio”????

Discussion:

A detailed and systematic health survey is required to affirm that IBF water caused less gastrointestinal illness. The same comment applies to the conclusion.

The authors could discuss what are the possible sources of fecal contamination in this spring water?

Reviewer 4 Report

The paper discusses a low cost technology that can better suit local rural communities due both lower costs (capex and opex) and also due to local tools/labor and materials. These characteristics are relevant for its environmental sustainability and it is also socially responsible.

The written English is quite straightforward to follow and mainly objective. I leave only small suggestions/corrections:

1 - line 323: replace “August …” by “…August 2022 …”;

2 - Line 324: replace “… dropped by two” by “…”only dropped by two…”;

Please, formulate hipotesis for this lower efficiency (2 log and not 2.5 log).

The IBF flow rate was the same during high rains, maintaining the residence time of about 19h? Some rain entered directly to the IBF? The biofilm layer was disrupted partially? Why? Temperature influence?

This episodic IBF lower performance can be critical (for water users) and should be investigated further here or future work. Please develop this type of discussion in present paper.

Around October 2022 the efficiency is even lower (last point in figure 3B). Why is that?

3 - line 199 (Figure2) consider adding the height of each layer, including water level;

4- line 282 (Figure 3) uses inches for rain. Consider using always IS units. The same for figure 1 (line 130) where the scale is in feets and could be in meters.

5 Why the precipitation data is not given in figures 4A and 4B? As given in figure 3.

Round 2

Reviewer 1 Report

This manuscript could be published in this form.

Reviewer 2 Report

The paper can be accepted after minor revision 

The authors should add XRF analysis for the filter materials. 

Reviewer 3 Report

The revised manuscript is suitable for publication .

Reviewer 4 Report

Replies to my previous comments are complete and now the paper is more complete and with less "mall gaps/mistakes". Thus, I support its publication as it is or with smaller changes that are neededand recommended by other reviewers or editors.